# Chitinase Signature in the Plasticity of Neurodegenerative Diseases

**DOI:** 10.3390/ijms24076301

**Published:** 2023-03-27

**Authors:** Cristina Russo, Maria Stella Valle, Antonino Casabona, Lucia Malaguarnera

**Affiliations:** 1Section of Pathology, Department of Biomedical and Biotechnological Sciences, School of Medicine, University of Catania, 95123 Catania, Italy; 2Laboratory of Neuro-Biomechanics, Section of Physiology, Department of Biomedical and Biotechnological Sciences, School of Medicine, University of Catania, 95123 Catania, Italy

**Keywords:** Alzheimer’s disease, Parkinson’s disease, amyotrophic lateral sclerosis, multiple sclerosis, glycohydrolase family 18, neuroinflammation, Chitotriosidase, chitinase-3-like 1

## Abstract

Several reports have pointed out that Chitinases are expressed and secreted by various cell types of central nervous system (CNS), including activated microglia and astrocytes. These cells play a key role in neuroinflammation and in the pathogenesis of many neurodegenerative disorders. Increased levels of Chitinases, in particular Chitotriosidase (CHIT-1) and chitinase-3-like protein 1 (CHI3L1), have been found increased in several neurodegenerative disorders. Although having important biological roles in inflammation, to date, the molecular mechanisms of Chitinase involvement in the pathogenesis of neurodegenerative disorders is not well-elucidated. Several studies showed that some Chitinases could be assumed as markers for diagnosis, prognosis, activity, and severity of a disease and therefore can be helpful in the choice of treatment. However, some studies showed controversial results. This review will discuss the potential of Chitinases in the pathogenesis of some neurodegenerative disorders, such as Alzheimer’s disease, Parkinson’s disease, amyotrophic lateral sclerosis, and multiple sclerosis, to understand their role as distinctive biomarkers of neuronal cell activity during neuroinflammatory processes. Knowledge of the role of Chitinases in neuronal cell activation could allow for the development of new methodologies for downregulating neuroinflammation and consequently for diminishing negative neurological disease outcomes.

## 1. Introduction

Neurodegenerative disorders (NDDs) include various pathological conditions, such as Alzheimer’s disease (AD), Parkinson’s disease (PD), dementia with Lewy bodies (DLB), amyotrophic lateral sclerosis (ALS), and multiple sclerosis (MS) [1]. NDDs can be classified as pyramidal and extrapyramidal movement disorders that cause either increased movements or reduced or slow movements [2]. In these different NDDS, there is a specific contribution of glial dysfunction, causing cognitive, psychomotor, and behavioral impairment, as in physiological conditions, glia are key factors in protecting the brain, furnishing nutrients and structural support for neurons in the central nervous system (CNS). Various factors encompassing genetic predisposition as well as environmental and metabolic factors contribute to NDD progression. Moreover, oxidative stress, mitochondrial dysfunction, calcium (Ca^2+^) influx, glutamate toxicity, proteolytic stress, protein aggregation, neuroinflammation, and neuronal death are the main molecular and cellular abnormalities associated with NDDs as a whole [3]. In many NDD microglia activations, synaptic and glial dysfunctions are the initial pathological event. Therefore, synaptic and glial dysfunctions and changes of networks between these cells characterize NDDs [4]. An accurate assessment of synaptic damage and glial dysfunction is indispensable to identify the physiopathology of neurodegeneration, to track disease evolution, and to obtain prognostic data. The diagnosis of neurodegenerative disorders is frequently hard because of the paucity of diagnostic tools, common pathological indicators, and comorbidities. Chitinases have been differently associated with neurologic disorder physiopathology hallmarks. Mammalian Chitinases belong mainly to the glycohydrolase family 18 [5]. These enzymes hydrolyze the glycosidic bond of chitin, an important structural constituent of fungi, nematodes, and arthropods [5]. Chitin is composed of linear β-1,4-linked *N*-acetylglucosamine (GlcNAc) units. Since many plants and animals employ Chitinases in their immune defense against chitin-containing pathogens, Chitinases are ubiquitous in nature [5]. The human genome encodes several Chitinase enzymes, able to degrade chitin encountered through consumption of breathing. Chitinases participate in nutrition, parasitism, morphogenesis, and immunity [6]. In humans, Chitinases have been distinguished in endochitinases, exochitinases, and chitobiases. These three classes are grouped on the basis of their catalytically active sites and action. Chitinases include members both with and without glycohydrolase enzymatic activity against chitin. Chitotriosidase (CHIT-1) and chitinase acid [7] are true Chitinases possessing chitinolytic activity to degrade chitin [5]. Inactive Chitinases lacking enzymatic activity, totally or partially, are termed chitinase-like proteins (CLP) or chilectins (ChiLs) [5]. Nevertheless, these groups of enzymes have well-maintained active-site carbohydrate binding and, therefore, are involved in several regulatory functions [5]. To date, four ChiLs have been identified in humans, termed chitinase-3-like 1 (CHI3L1), chitinase-3-like 2 (CHI3L2), oviductal glycoprotein 1 (OVGP1), and chitinase domain-containing protein-1 or stabilin-1 [6]. They are deficient in chitin-hydrolyzing activity but possess cytokine-like and growth factor-like properties. CHI3L1 (also known as YKL40, or human cartilage glycoprotein-39, HC-gp39), shows enzymatic activity despite the retention and conservation of the substrate-binding cleft of the Chitinases [6]. ChiLs are involved in various processes such as tissue remodeling and injury [8]. The human Chitinase family has been extensively exanimated due to their excessive secretion into the serum or overexpression in tissues that are chronically inflamed [5]. Elevated levels of true Chitinases have been reported in a variety of diseases including infections, chronic inflammation, degenerative disorders, and cancer [5]. Active true Chitinase mammalian ChiLs are involved in immunomodulation, a pathological condition with an inflammatory pathogenesis [5,6,9,10]. Several investigations detected the dynamic changes of Chitinases in cerebrospinal fluid (CSF) as an expression of microglia activation and synaptic and glial dysfunctions, possibly related to cognitive impairment. CHIT-1 and CHI3L1 have been the most investigated from the clinical and biological perspectives of a variety of inflammation-prone brain diseases. Among the Chitinase family, they have been recognized as markers of neuroinflammation in a variety of neurological diseases [11,12,13]. Nevertheless, another Chitinase, CHI3L2, has been involved in neuroinflammation pathogenesis of neurodegenerative diseases [14,15]. In this review, we summarize the available data on Chitinase expression patterns as well as on their contribution in glial cells and plasticity and other less well-characterized roles. Moreover, we propose a mechanism for their involvement in synaptic damage and neurodegeneration and assess their potential as CSF biomarkers for neurodegenerative diseases.

## 2. Chitin and Chitinase Induction in Alzheimer’s Disease (AD)

AD is one of the most common neurodegenerative diseases worldwide [16]. It is a progressive neurodegenerative condition resulting in the regression of intellectual capabilities, memory loss, and spatial disorientation due to neuronal cell damage in higher brain centers [16]. The pathogenesis of AD mainly includes deposition of amyloid-beta (Aβ) protein and deposition of neurofibrillary tangles (NFTs) of tau proteins, which increase inflammatory response with accumulation of reactive oxygen species (ROS). Moreover, hormonal disorders, genetic predisposition, mitochondrial dysfunction, lack of neurotropic factors, metal ion dynamic equilibrium disorder, calcium toxicity, and acetylcholine (ACh) deficiency [17] contribute to the development of AD [16]. Originally, it was thought that chitin was absent in vertebrates, including humans. Only in the last decade has chitin been recognized as an element present also in humans. Basically, the increase of light chitin debris takes place in subjects with low Chitinase activity, as a result of peripheral fungal infections, or when rapid synthesis of hyaluronan occurs. Some studies have shown that chitin, being toxic for neurons, can also be involved in AD pathogenesis. Chitin is an insoluble molecule and a substrate for glycan–protein interactions. In the brain, chitin could facilitate nucleation of amyloid proteins resulting in AD onset [18]. In fact, elevated levels of chitin have been found in plasma, in CSF cells, and in CNS cells of AD patients [19]. When introduced, chitin has been found to exert both pro-inflammatory and anti-inflammatory effects, which are based on the size of the molecule [20]. Chitin molecules larger than 70 µm are named big chitin (BC) and are inert to the inflammatory response [21]. Nevertheless, if chitin molecules get degraded by mammalian Chitinase, they can generate small chitin (SC) and intermediate chitin (IC) molecules of a size minor than 40 µm and between 40 and 70 µm, respectively. SC induces an anti-inflammatory response via the production of interleukin 10 (IL-10) (Figure 1) [21]. IL-10, being an anti-inflammatory cytokine, inhibits the immune response. Its binding with its own receptors IL-10R1 and IL-10R2, triggering the JAK/STAT signaling pathway causes inhibition of pro-inflammatory cytokine production and phagocytosis inhibition [22,23]. Whereas IC stimulates a pro-inflammatory response via Toll–like receptor 2 (TLR2)- and nuclear factor κ-light chain of enhancer-activated B cell (NF-κB)-dependent pathways for tumor necrosis factor (TNF) production (Figure 1) [21]. A feature in the brain of AD patients is the presence of activated microglia and astroglia, releasing pro-inflammatory cytokines and chemokines [24]. The neuroinflammatory hypothesis of AD suggests that Aβ-induced microglia activation, and the consequent phagocytosis, lysosomal impairment, and NLR Family Pyrin Domain Containing 3 (NLRP3) inflammasome activation, affect microglia-induced neurotoxicity. It has been demonstrated that microglial cells are able to engulf chitin debris [24]. Next to their activation, both microglia and neurons produce GlcNAc polymers in the extracellular space that trigger neurotoxicity (Figure 1) [25]. It is also conceivable the amassing of IC adds a new element in microglia activation and neurotoxicity. Therefore, deposits of chitin in the brain of AD patients strongly indicate their contribution in eliciting the expression of Chitinases and the subsequent neuroinflammation in the pathogenesis of AD [11].

## 3. Chitinases and Activated Microglia in Neuroinflammation

Subsequent to the onset of AD, the massive formation of amyloid plaques and intracellular neurofibrillary tangles enclosing a misfolded phosphorylated tau protein induces synaptic dysfunction followed by axonal impairment and cognitive changes. While this protein-related process is ongoing, the immune system is highly responsive [26].

Misfolded and aggregated proteins, conceivably including IC, binding to pattern recognition receptors on microglia and astroglia [21,24,25], trigger neuroinflammatory response characterized by the release of inflammatory mediators, which contribute to the process of neurodegeneration and to disease progression [21,26,27,28].

Microglia are immunomodulatory cells playing a significant role in immune surveillance of the CNS. In the course of neuroinflammation, specific pro-inflammatory factors stimulate microglia. Microglia have phagocytic activity to remove infectious agents, neurofibrillary plaques, and damaged neurons. Microglia are generally considered the primary innate immune cells of the CNS. In response to many types of damage, microglia first change their morphology from ramified to amoeboid form, and then they migrate to the damaged cells and clear the debris of the dead cells by phagocytosis. Microglia recognize damaging stimuli and react by producing inflammatory cytokines and several chemokines [29]. By promoting the release of pro-inflammatory cytokines, microglia are involved in the innate response providing a rapid control of invading pathogens and influencing T and B cell activation. In the immune response, microglial cells act in different ways releasing pro-inflammatory molecules and acting as antigen-presenting cells.

Additionally, microglia control the interaction between neurons and astrocytes for the restoration and reorganization of damaged synapses. Microglia polarize in two different phenotypes: cytotoxic M1, pro-inflammatory, and cytoprotective M2 [30]. After the induction of TLR, microglia activation leads to proinflammatory mediator release such as TNF-α, interferon-gamma (IFN-γ), IL-1(IL-1β), IL-6, and IL-12. These cytokines increase oxidative stress and nitrogen free radicals [30], and they also promote neuroinflammation, which is the major pathological feature of neurodegenerative diseases [29]. In contrast, IL-4 and IL-10 promote the M2 phenotype (Figure 1). M2 microglial cells reduce brain inflammation, inhibiting M1 cytokines and other inflammatory mediators and exerting regenerative functions to restore homeostasis [31].

It has recently been reported that microglia display an overlapping functional state, shifting from one to the other depending on the activated pathways. Other phenotypes of microglia, such as M2a-like microglia, M2b and M2c, have been identified to be able to express both pro-inflammatory and anti-inflammatory markers [31]. In particular, the M2b phenotype, producing both inflammatory and restorative markers, would constitute an intermediate state. Whereas M2a-like microglia, stimulated with IL-4 and IL-13, induce arginase, chitinase 3-like 3 (or Ym1) gene, and CD206 [32]. These data suggest to deepen the role of CH3L3 in activated microglia during AD.

M2c suppresses the immune response upon IL-10, induces tissue remodeling, and restoration producing the transforming growth factor beta (TGF-β), CD206, and CD163 [33], followed by the resolution of the neuroinflammation. Thus, the M1-polarized phenotype promotes neuronal damage, whereas the M2-polarized phenotype is immunosuppressive and neuroprotective (Figure 1) [33].

Disturbance of microglial functional polarization is implicated in the pathogenesis of a wide variety of CNS disorders, including MS, stroke, AD, and other neurodegenerative diseases [30].

## 4. CHIT-1 and AD

Neurons and microglia exposed to GlcNAc and uridine diphosphate (UDP)-GlcNAc are able to form GlcNAc polymers, which display a significant neurotoxicity in vitro. The exposure of hippocampal cultures to the GlcNAc polymers induces synaptic impairment with decreased levels of syntaxin and synaptophysin. Hippocampal slices treated with GlcNAc/UDP-GlcNAc showed a clear reduction of long-term potentiation of excitatory synapses [25]. The increased levels of CHIT-1 expression and activity in microglia may be dependent on GlcNAc polymers. CHIT-1 is one of the most important active enzymes able to degrade chitin polymers [5], CHIT-1 is produced, stored, and secreted by activated macrophages; hence, it is highly involved in the development of the immune response and inflammatory processes [6].

Its enzymatic activity increases in a number of diseases associated with macrophage activation [5], such as Type 2 diabetes mellitus [9], atherosclerosis, and steatosis [34]. Several reports have shown that CHIT-1 activity is markedly elevated in AD [11,35,36].

It was previously reported that O_2_^−^ generation and cytokine production in macrophages of peripheral blood of AD and cerebrovascular dementia patients promote CHIT-1 activity. A massive production of ROS induces increased levels of lipid peroxidation that accelerated Aβ deposition in a transgenic mouse model of AD [11]. Nevertheless, to date, the data showing possible implications of CHIT-1 in the pathogenesis of AD are contradictory. These conflicting data may be related to the different stages of the disease. In fact, it has been observed that in clinically advanced stages of the disease, the levels of CHIT-1 activity did not show significant differences compared with controls [37]. In contrast, Mattsson et al. [38] found higher CHIT-1 levels in CSF samples of the AD group, when compared with controls and patients with stable mild cognitive impairment (sMCI) [38]. In rat models of AD, Yu et al. [39] reported that CHIT-1 exerts potential protection through microglial polarization and reduction of β-amyloid (Aβ) oligomers [39]. They demonstrated that CHIT-1 promotes brain inflammation via the HDAC3/NF-κB p65 pathway, contributing to improvement of cognitive impairment and AD progression (Figure 1 and Figure 2) [39].

## 5. CHI3L1 and AD

As mentioned before, CHI3L1 belongs to chitinase-like lectins [40], which are related to the glycohydrolase family 18. CHI3L1 has no enzymatic activity because of a substitution of the catalytic glutamic acid residue with leucine [41]. CHI3L1 is produced by a large quantity of cells, including neutrophils, monocytes/macrophages, monocyte-derived dendritic cells, osteoclasts, chondrocytes, fibroblasts, endothelial cells, and vascular smooth muscle cells [40,42]. The elevated level of CHI3L1 in inflammation sites influences its ability to regulate cell proliferation, adhesion, migration, and activation of the extracellular matrix assembly. Moreover, CHI3L1, inducing inflammatory mediators, including Chemokine ligand 2 (CCL2), Chemokine (C-X-C motif) ligand 2 (CXCL2), and marrow matrix metalloproteinase 9 (MMP-9), acts as a pro-inflammatory biomarker (Figure 1) [43,44]. Induction of CHI3L1 has been reported in patients suffering from a large number of diseases, especially inflammation-related illnesses [44], and autoimmune disorders [6]. Elevated plasma levels of CHI3L1 have been found in atherosclerosis, coronary artery disease, and neurodegenerative conditions. Variations of CHI3L1 levels have been recognized between AD patients and healthy elderly subjects [45]. CHI3L1 concentrations result as consistently high in the dementia phase of AD [46,47,48]. It has been also reported that higher CHI3L1 levels mainly reflect a response to both total tau and phosphorylated tau levels in the pre-dementia AD group. Therefore, CHI3L1 is an important parameter to detect early pathophysiological changes in connection with the neurodegenerative process [49]. However, in the literature, discordant observations are often reported. Gispert et al. [50], investigating CSF CHI3L1 levels between normal, preclinical, and mild dementia, reported that CSF CHI3L1 is associated with a cerebral structural signature distinct from that linked to phosphorylated tau neurodegeneration at the earliest stages of cognitive decline due to AD [49]. Other studies reported no significant differences in CSF CHI3L1 levels between sMCI and MCI-AD patients [51] and between AD and MCI patients. Others found higher CHI3L1 levels in MCI and AD patients compared to controls, suggesting that this protein could have a strong importance in detecting cognitively normal individuals from MCI and AD patients. In addition, it has been suggested that CHI3L1 levels may be associated with disease progression [52]. Nevertheless, a study demonstrated increased CHI3L1 CSF levels in AD patients compared with MCI patients who developed vascular dementia, but not in MCI-AD patients (Figure 2). Moreover, in AD patients with a significant correlation between time points of cortical damage, total tau and CHI3L1 levels were constant [53]. In another investigation in MCI and AD patients was observed a nonlinear association between gray matter volume and CHI3L1 levels in inferior and lateral temporal areas spreading to the supramarginal gyrus, insula, inferior frontal cortex, and cerebellum [54]. Additionally, in AD patients, CHI3L1 levels were also associated with decreased cortical thickness [55].

One of the neuropathological features of AD is the crucial role played by microglial cells. Since microglia express various pro-inflammatory cytokines at molecular and cellular levels, they are crucial for the immune response in the brain. In brain tissue, CHI3L1 is expressed in microglia, infiltrating macrophages and astrocytes [56]. The higher CHI3L1 levels in the CSF and in the plasma of AD patients compared with healthy controls reflect the inflammatory evolution of the disease. The expression of CHI3L1 is modulated by pro-inflammatory cytokines, including IL-6, IFN-γ, IL-1β, and TNF-α [10] Its regulation by IL-6 and TNF-α requires sustained activation of NF-κB. Additionally, CHI3L1 activates the protein kinase B and phosphoinositide-3 kinase signaling pathways and exerts an anti-apoptotic function [57]. CHI3L1 induces IL-4Rα expression and phosphorylation of STAT6, affecting microglial M2 polarization. In fact, in a knockdown model, it has been observed that CHI3L1 decreases the expression level of IL-4Rα, phospho-JAK1, phospho-JAK3, and phospho-STAT6 blocking microglial M2 polarization (Figure 1) [58]. In contrast, it has been found that CHI3L1 immuno-reactivity is mainly expressed in astrocytes than microglial cells in the frontal cortex of AD patients [59]. Reactive astrocytes are involved in the pathophysiology of AD [60] and are among the most important cellular mediators in the neuroinflammatory response observed in AD [16]. Transcriptomic analyses proved that reactive astrocytes can gain multiple molecular phenotypes in AD, suggesting that they respond differently to AD-related brain processes [47]. Reactive astrocyte CHI3L1 levels are consistently elevated in the dementia phase of AD [48]. Post-mortem investigations show that both Aβ and tau pathologies are linked with astrocyte reactivity [61]. Moreover, CHI3L1 levels have been strictly related to the levels of CSF inflammation-related proteins [48]. Reactive astrocytes overexpress CHI3L1, which is released into the extracellular compartment, and so, they can be measured in the CSF. Therefore, CHI3L1 is increasingly recognized as a reactive astrocyte biomarker in AD [47,48]. CHI3L1 mediates the effects of Aβ and tau pathologies on hippocampal atrophy and cognitive impairment. It has been demonstrated that, in CSF, the CHI3L1 levels were linked with tau accumulation—but not Aβ—in brain regions typically affected by AD-related tau pathology. This result suggests that CHI3L1 levels in the CSF mirror an astrocyte response to tau tangles accumulation in AD [49]. Indeed, post-mortem investigations showed astrocyte overexpression of CHI3L1 in AD and other tauopathies [55], (e.g., Pick’s disease, corticobasal degeneration, and progressive supranuclear palsy) [62]. More recently, it was reported that in the brain of AD patients CHI3L1 is co-localized with glial fibrillary acidic protein (GFAP) (an astrocyte marker) [63]. CSF CHI3L1 levels correlate with other AD-related inflammatory markers, such as IL-8 [64] and CXCL1 [65], suggesting that CHI3L1, activating the innate immune response, can be directly involved in the inflammatory response [47]. In addition, CHI3L1 has been linked with neuroinflammatory proteins associated with AD progression, such as metalloproteinase-10 (MMP-10) [66], CX3CL1 [67], eukaryotic translation initiation factor 4E-binding protein 1 (also known as 4E-BP1) [68], and CSF-1 (Figure 1) [69]. In conclusion, the increased CHI3L1 levels in the CSF of AD patients demonstrate their significant role in neuroinflammation and in the pathophysiology of the disease. CHI3L1 levels may be important for the evaluation of cerebral inflammatory activity in AD patients, being increased in the CSF of MCI-AD patients compared with stable MCI, CHI3L1 could be used as a reliable biomarker for the prognosis of MCI and the likelihood of progression to AD (Figure 2) [70].

## 6. Chitinases, Parkinson’s Disease (PD), and Dementia with Lewy Bodies (DLB)

PD is the second most common neurodegenerative disease worldwide adversely affecting quality of life. It is a complex and progressive disease characterized by a variety of clinical manifestations including tremors, rigidity, bradykinesia, postural instability, confusion, and depression [71]. The pathological features of PD are the significant loss of dopaminergic neurons in the substantia nigra pars compacta and the presence of intraneuronal proteinaceous cytoplasmic inclusions named Lewy bodies. The increase of Lewy bodies in nerve cells causes another neurodegenerative disorder, dementia with Lewy bodies (DLB) [72].

The progressive neuronal loss and PD and of DLB symptoms have been linked to α-synuclein (αSyn) aggregates, which result in axonal and neuronal damage [73].

Other than αSyn aggregate detection, other pathophysiological events are involved in neurodegeneration. PD and AD exert common neurodegenerative processes [74]. A great percentage of PD patients displays cognitive decline or even dementia during the disease course and an AD-like pathology with the presence of Aβ plaques and tau protein-containing neurofibrillary tangles [75]. The αSyn aggregates spreading through the brain, induced by Aβ deposits and phosphorylated tau, can lead to neuronal loss and correlate with cognitive and motor decline [75]. Moreover, αSyn aggregates interact with several cells of the immune system including microglia [76], triggering neuroinflammation. In PD, microglial activation is involved in the initiation and progression of the disease through secretion of pro-inflammatory cytokines and reactive oxygen species [77]. As PD progresses, through the PET assay, an early inflammation in the brainstem before cortical spreading is detected. In the initial stages of PD, an elevated microglial activation appears to be widespread in the substantia nigra [78]. Although CHI3L1 is involved in synaptic degeneration, glial activation, inflammation, and AD co-pathology, its CSF levels do not differ significantly between PD patients and healthy controls [79]. To explain this result, it has been hypothesized that the neurodegeneration in PD initially involves a smaller number of neurons than AD. Hence, neuroinflammation and synaptic impairment may not be widespread enough to induce CHI3L1 CSF.

Nevertheless, the level of CHI3L1 increases in PD groups showing MCI, suggesting that CHI3L1 CSF levels correlate to glial response, axonal damage, and worse cognitive performance. These data outline the importance of the use of ongoing follow-ups to predict cognitive decline in PD [79]. As seen in AD, astrocytes play an important role in neurodegeneration of PD too [80]. Astrocytes participate in innate immune responses in the CNS and may also play a role in adaptive immunity [81]. αSyn aggregates released by neurons can be internalized by adjacent astrocytes, forming glial inclusion bodies, which can induce changes in gene expression of astroglia, activating an immune response that facilitates neurodegeneration [82,83]. In addition to the direct effect of αSyn on microglia, astrocytes can mediate microglial activation. This implies that astrocytes control the timing of microglial activation, leading to accelerated neuronal damage and disease progression [84]. Glial inflammatory responses are related to the extent of deposition of αSyn aggregates. αSyn induces Lewy-like inclusion bodies and apoptotic changes in receiving neurons. Therefore, excess of neuronal αSyn is transferred and accumulated in glia, becoming an important stimulus of astroglial inflammatory responses that induces the formation of pathological inclusions and degenerative changes [85]. Besides inflammatory changes, αSyn accumulation in astrocytes causes autophagy defects, resulting in more inclusions [86]. Studies on the correlation between αSyn aggregates and CHI3L1 production are limited. In an investigation of Morenas-Rodríguez et al. [87], it was found that the glial marker CHI3L1 was not increased in CSF in the prodromal phase and in the later stages of dementia with Lewy bodies (DLB). In contrast, CHI3L1 levels increased in CSF of DLB patients only in the presence of AD co-morbidity [87]. These findings were in agreement with other findings in PD and DLB [88]. The unchanged levels of CSF CHI3L1 in synucleinopathies suggest a failure of astroglial activation following αSyn amassing. The difference between αSyn and tau-related neurodegenerative dementias shows that CHI3L1 is strictly involved in tau-related disorders. Therefore, the results for which CHI3L1 levels correlate with total tau and phosphorylated tau levels in DLB groups, suggest that DLB in combination with AD comorbidities promotes astrocytic activation in response to pathologic proteins [87]. Moreover, it is possible that the αSyn inclusions in astrocytes of DLB may influence the astrocytic response toward neurodegeneration compared to tauopathies such as AD and frontotemporal lobar degeneration (FTLD) [89]. CSF CHI3L1 was also unchanged in prodromal DLB, demonstrating that this glial marker does not change early in the course of the disease (Figure 2). In the same study, the CHI3L1 level of CSF was investigated in DLB and prodromal DLB (prodDLB) patients compared with AD patients. Levels of CHI3L1 between DLB and prodDLB patients showed no significant differences. DLB and prodDLB patients had lower CHI3L1 levels than AD patients. However, CHI3L1, as well as phosphorylated tau and phosphorylated tau/Aβ and total tau/Aβ combinations, have a predictive value of cognitive decline in PD during follow-up, as revealed by correlating CSF levels with cognitive measurements. Therefore, DLB and AD show different models of glial activation markers in the CSF [90].

CHI3L1 is only increased in DLB when there is an underlying AD pathology and, contrary to AD, levels of CHI3L1 are not elevated in prodromal stages. Together, these findings suggest a different glial activation model between DLB and AD. This finding needs further studies to clarify CHI3L1 involvement in synaptic degeneration and glial activation, as well as its differential role in the innate immune response in DLB and its impact on pathogenesis and disease progression.

## 7. Chitinases and Amyotrophic Lateral Sclerosis (ALS)

ALS is a devastating and incurable neurodegenerative disorder, in which the selective neurodegeneration and loss of the upper and lower motor neurons causes spasticity and progressive disability. Prognosis is poor, and death comes within 3–5 years, mainly caused by respiratory failure. A wide clinical variety of ALS with numerous subtypes and a broad phenotypic heterogeneity has been identified [91]. Likewise, the early symptoms differ significantly between patients ranging from foot drop to dysarthria. Several subjects present behavioral alterations and cognitive impairment. Some ALS patients meet the criteria for frontotemporal dementia (FTD) [92]. The disease may be genetic, familial ALS, or sporadic (SALS). ALS is characterized by reactive gliosis and neuroinflammatory activation that lead to disease progression. It has been postulated that ALS is triggered mainly by genetic risk factors and to a lesser extent by environmental determinants [93]. Environmental factors include smoking, exaggerated physical activity, and recurrent head injury [94]. The main pathological hallmark of ALS is the neuronal inclusion of TAR DNA-binding protein 43 (TDP-43) in the brain and spinal cord of patients [95]. Cytoplasmic TDP-43 aggregation generates neurotoxicity through dysregulation of nuclear RNA processing, and a sequence of aberrant processes, such as atypical stress granule formation, mitochondrial dysfunction, altered translation, proteasomal dysfunction, cellular stress, and reduced autophagy [96]. The pathogenesis of TDP-43 mutation in ALS has been observed in various animal models such as mice [97], rats [98], and monkeys [99], as well as cultured human motor neurons differentiated from reprogrammed stem cells [100]. TDP-43 proteinopathies have also been observed in patients with ALS or frontotemporal lobar degeneration (FTLD) without pathogenic mutation of TDP-43 [101]. Altered TDP-43 expression may unbalance the pathways resulting in severe outcomes, including neuronal death [102]. Astrocytes may play an important role in TDP-43 pathogenesis. In response to TDP-43 mutation, neurotoxic effects generate changes in the expression of astrocytic receptors and transporters; the coupling of gap junctions; the release and metabolism of astrocytic transmitters; and production of chemokines, cytokines, and free radicals [103]. Overexpression of TDP-43 promotes activation of astrocytes and microglia in the spinal cord. Besides TDP-43, there are several other commonly known ALS genes, such as C9ORF72 and superoxide dismutase 1, fused in sarcoma [104]. Reactive astrocytes are able to secrete the neurotoxic factor Lipocalin 2 (Lcn2) [103]. Lcn2 is a lipocalin sensitizing neurons to Aβ-toxicity, exerting selective neurotoxicity [103,105].

Neuroinflammation, featured by glial cell activation and activation of other cells of the innate and adaptive immune system, is a key component of the non-cell autonomous neurodegeneration in ALS [106]. Among the numerous inflammatory markers that have been detected in ALS Chitinases, CHIT-1, CHI3L1, and CHI3L2 are those which demonstrated better efficiency at diagnostic and prognostic levels [107]. Cross-sectional analysis showed elevated CSF levels and CHI3L1 and CHI3L2 concentrations, as well as CHIT-1 levels and activity in symptomatic ALS compared with both at-risk subjects and healthy controls [14,108]. Longitudinal studies delineate the chronological profile of Chitinase response in ALS. There is a gradual rise in CSF CHIT-1 protein in at-risk subjects during the pre-symptomatic period, followed by a rapid increase in CHIT-1 levels and activity between the late pre-symptomatic and early symptomatic stages of the disease among phenoconverters, followed by a progressive increase in CHIT-1 levels during the symptomatic period [109]. Particularly, the increase in CHIT-1 levels in at-risk individuals over time was greater than would be estimated for age based on cross-sectional analysis of the association between CHIT-1 and age [109]. Studies on frontotemporal dementia, ALS, and AD patients reported that CHIT-1 expression reflects microglial activity and is associated with the rate of disease progression in ALS [110]. Since CHIT-1 is primarily expressed by myeloid lineage cells, this evidence suggests that microglia are rather quiescent in the period previous to transition to symptomatic ALS. Changes in the levels of CHIT-1 protein and activity during the transition of the pathological manifestation reflect a late pre-symptomatic or early post-symptomatic occurrence. Nevertheless, it has not yet been clarified how the timing of the peri-symptomatic increase in CHIT-1 protein and activity are correlated with the increased expression of TDP-43. It remains undefined whether increased CHIT-1 reflects a “physiological” response to cytoplasmic TDP-43 aggregation, or whether increased microglial activity and CHIT-1 overexpression are events preceding neurodegeneration. Nevertheless, it has been found that homozygous carriers of the CHIT-1 polymorphism, which generate the reduction of CHIT-1 expression, do not exhibit decreased ALS severity, indicating that CHIT-1 is a neuroinflammatory marker rather than being actively involved in ALS pathogenesis (Figure 2) [110,111].

Astrocyte activity increases rapidly in the peri-symptomatic period of ALS before decreasing over the course of the symptomatic period. Reactive astrocytes are the cells preferentially producing CHI3L1 as demonstrated by histopathological analysis in ALS [112], in which it was observed that CHI3L1, but not CHIT-1 or CHI3L2, is a marker more closely associated with clinical features of central dysfunction denoted by upper motor neurons. This result supports the idea that CHI3L1 has a prognostic value for cognitive and behavioral impairments in patients with ALS [14,112]. In addition, CHI3L1 was found expressed in the anterior horn of the spinal cord in patients with sporadic ALS [113]. CHI3L1 and CHI3L2 expressions have been found to be significantly increased in the motor cortex of sporadic ALS patients compared with neurologically healthy controls, and their expression levels were correlated with the survival time from date of onset [14,108]. In addition, CSF CHI3L1 levels correlate significantly with biomarkers of neurodegeneration such as the neurofilament light chain and the phosphorylated neurofilament heavy chain protein [114].

CHI3L1 displays an immunoreactivity limited to glial fibrillary acidic protein (GFAP) -positive astrocytes in the frontal cortex and spinal cord of patients with ALS [113]. It has been reported that CHIT-1, CHI3L1, and CHI3L2 levels can differentiate patients with ALS from ALS mimics, which are patients affected by motor weakness without ALS [14]. The increased CSF levels of CHIT-1, CHI3L1, and CHI3L2 correlate with disease progression grades of ALS patients. In particular, CSF levels of CHIT-1 and CHI3L1 predict a shorter survival [107]. CHIT-1 is primarily expressed by cells of myeloid lineage, and CHI3L1 is produced by reactive astrocytes and microglia [115]. It has been found that CHIT-1 immunostaining co-localizes with the microglial marker Iba1 and phagocytic activity marker CD68 in the spinal cord of patients with ALS (Figure 2) [116].

Overall, these investigations validate that Chitinases, regarded as valuable candidates for evaluating the course of ALS, can be promising new targets as pharmacodynamic biomarkers for neuroinflammation-focused ALS therapy [12,115].

## 8. Chitinases and Multiple Sclerosis (MS) 

MS is an autoimmune-mediated demyelinating disease of the CNS. It is the most common neurodegenerative disease affecting young adults [117]. Usually, symptoms consist of fatigue, depression, and failure of motor and cognitive functions, as well as visual, bowel, and bladder dysfunctions [118]. Subjects with primary progressive multiple sclerosis experience the aggravation of neurological function, progressive disability, and worsening physical and psychosocial quality of life [119].

MS patients can experience a long-term benign course to a highly active disease. Several patients present a widely variable disease course featured by relapse periods (rapid disability with worsening symptom) followed by periods of remission (long-term benign course). For many people, relapsing-remitting MS (RRMS) precedes secondary progressive MS, in which disability gets steadily worse. MS onset has been recognized as a multifaceted interplay of genetic, environmental, and lifestyle risk factors [120]. The environmental and lifestyle risk factors for MS onset include cigarette smoking [121], youth obesity [122], and Epstein–Barr virus infection [123].

To date, molecular alterations driving MS susceptibility have not been fully resolved. Important factors of pathophysiology of MS embrace a massive migration of leukocytes into the CNS parenchyma due to the impairment of blood–brain barrier (BBB) integrity, which generates the inflammatory response, oligodendrocyte death, demyelination, gliosis, and neurodegeneration [124]. Both microglia and astrocyte dysfunctions can be involved in MS pathology. Microglia and astrocyte accumulate at sites of active demyelination and neurodegeneration in the brain of MS and are believed to play a central role in the process of the disease. In the early stages of demyelination and neurodegeneration, active lesions contain microglia with a pro-inflammatory phenotype, expressing molecules involved in phagocytosis, oxidative damage, in the presentation of the antigen, and in the co-stimulation of T cells. In the subsequent stages, microglia in active lesions take in the intermediate phenotype between pro- and anti-inflammatory activation [125] Astrocytes promote leukocyte recruitment and produce B cell-activating factors when proximal to lesion area, exacerbating leukocyte activation [126]. Hence, they participate directly in the innate immune response, contributing to early MS onset. In addition, astrocytes take on an active state and phagocytose myelin during the earliest stages of lesion formation. Astrocytes are able to activate and form an astroglial scar within chronic silent late-stage lesions [126]. On the other hand, inflammation impairs astrocytes, further contributing to injury by deferring microglia and oligodendrocyte progenitor recruitment to the lesion level, and finally impeding myelin debris removal and demyelination [127]. Astrocytes can disconnect from the vasculature in MS, as evidenced by reduced GFAP immunoreactivity around blood vessels [126].

Chitinase expression exactly reflects microglia and astrocyte dysfunctions. CHI3L1 has attracted growing attention as a biomarker of neuroinflammation and microglial activation and reactive astrocytes, two signs both particularly evident and relevant in MS pathogenesis [128]. The role of CHI3L1 as a candidate biomarker of clinical conversion to MS and development of disability was initially reported by Comabella et al. [129].

It has been shown that CSF and serum CHI3L1 levels increase with the disease stage and clinically isolated syndrome and that CHI3L1 converted more rapidly to RRMS patients. In contrast, it was found that CSF CHI3L2 levels were lower in progressive MS (PMS) than in RRMS patients (Figure 2) [130,131]. Hence, the CSF CHI3L1/CHI3L2 ratio might help to define the MS disease stage and have a prognostic value in clinically isolated syndrome (CIS) [130].

More recent cross-sectional studies have demonstrated a significant association between CHI3L1 levels in CSF and Expanded Disability Status Scale, suggesting that there were high levels of CHI3L1 in the CSF of MS patients, and there was a significant correlation between CHI3L1 and oligoclonal bands. Therefore, CHI3L1 has been considered a promising diagnostic marker of MS [132].

So far, only few investigations have indicated CHIT-1 as a prognostic factor for early identification of patients presenting with CIS at high risk for conversion to MS or to recognized RRMS patients at high risk for progression [133,134].

## 9. Conclusions

From the many experimental reports, it has been established that Chitinases are molecules that can be used as biomarkers to monitor the progress of the different stages of neurodegenerative diseases characterized by neuroinflammation, microglia activation, and astrocytes. The common denominator in the degenerative pathologies analyzed in this review is the activation of microglia and astrocytes resulting from the accumulation of molecular aggregates of various nature, such as β-amyloid plaques and tau protein-containing neurofibrillary tangles, GlcNAc polymers, observed in AD, αSyn aggregates in PD, and DLB and TDP-43 in ALS (Figure 2). Therefore, the accumulation of Chitinases in neurodegenerative diseases, here examined, may have a deeper molecular significance. We still have no critical knowledge of the molecular pathways in the expression of Chitinase in brain cells, nor is the nature of molecular aggregates that induce their activation fully known. Among the identified proteins that play a role in the activation of microglia or astrocytes, such as tau or αSyn proteins, it is not excluded that accumulation of intermediate chitin can play a not inconsiderable role in the activation of neuronal cells and in neuroinflammation. Therefore, through the analysis conducted in this review, we are only able to understand a small fraction of the overall picture known so far. Maintaining balance in the plasticity of the cells of the nervous system is mandatory to delay the devastating aspects of the pathologies under consideration. Acquiring a greater understanding of the mechanisms by which Chitinases are regulated can provide new tools to modulate their action.

## 10. Future Prospects

Studies on the mechanisms of action and the signal pathways used by Chitinase are necessary to improve understanding the intermediate chitin accumulation role and the Chitinase action, in the nervous system under physiological and pathological conditions. The results gathered could be beneficial not only for early diagnosis of neurodegenerative diseases, but also for valid treatments in order to preserve patients from the risk of severe cognitive decline.

## Figures and Tables

**Figure 1 ijms-24-06301-f001:**
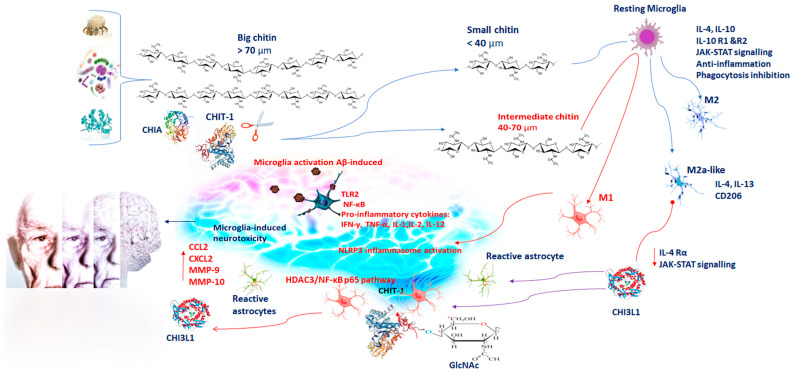
Chitin has both pro-inflammatory and anti-inflammatory effects on the basis of its molecular size: BC (>70 µm) is inert to the inflammatory response; SC (<40 µm) exerts an anti-inflammatory response via IL-10 production. IL-10, binding with its own receptors IL-10R1 and IL-10R2, triggers the JAK/STAT signaling pathway, resulting in pro-inflammatory cytokine and phagocytosis inhibition; IC (40–70 µm) induces pro-inflammatory responses via TLR2- and NF-κB-dependent pathways for IFN-γ, TNF-α, IL1, IL2, and IL12 production. Microglial cells engulfing chitin debris, GlcNAc polymers, and Aβ induce NLRP3 inflammasome activation and neurotoxicity. CHIT-1 and CHIA are the most important active enzymes able to degrade chitin polymers. CHIT-1 modulates neuroinflammation via the HDAC3/NF-κB p65 pathway contributing to AD progression. Microglia polarize in M1 and M2, displaying pro-inflammatory and anti-inflammatory phenotypes, respectively. The M1-polarized phenotype promotes neuronal damage, whereas the M2-polarized phenotype is immunosuppressive and neuroprotective. M2a-like microglia induce CHI3L1, IL13, and CD206. CHI3L1 modulates IL-4Rα expression and STAT6 phosphorylation, affecting M2 polarization. CHI3L1 is expressed in microglia, infiltrating macrophages and astrocytes. The expression of CHI3L1 is induced by pro-inflammatory cytokines, including IL-6, IFN-γ, IL-1β, and TNF-α. Its regulation by IL-6 and TNF-α needs NF-κB activation. CHI3L1 is induced by inflammatory proteins, such as MMP-10, CX3CL1, and 4E-BP1, and therefore, its activity facilitates AD neuroinflammation. Blue arrows represent induction; red arrows represent inhibition. Abbreviation= BC, IC, SC: big, intermediate, small, chitin; IL: interleukin; TLR2: Toll–like receptor 2; NF-κB: nuclear factor kappa-light-chain-enhancer of activated B cells; IFN-γ: interferon-gamma; TNF-α: tumor necrosis factor-α; GlcNAc: β-1,4-linked *N*-acetylglucosamine; Aβ: beta-amyloid protein; NLRP3: NLR Family Pyrin Domain Containing 3; CHIT-1: Chitotriosidase; CHIA: chitinase acid; AD: Alzheimer’s disease; CHI3L1: chitinase-3-like protein 1; MMP: marrow matrix metalloproteinase; CXCL: Chemokine family; 4E-BP1: eukaryotic translation initiation factor 4E-binding protein 1.

**Figure 2 ijms-24-06301-f002:**
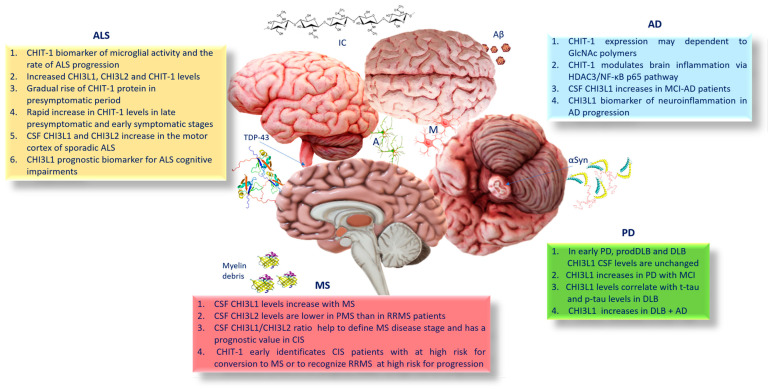
Schematic representation of the chronological profiles of Chitinases in AD, PD, ALS, and progression. Abbreviation = AD: Alzheimer’s disease; PD: Parkinson’s disease; ALS: amyotrophic lateral sclerosis; MS: multiple sclerosis; CSF: cerebrospinal fluid; CHI3L1: chitinase-3-like protein 1; CHI3L2: chitinase-3-like 2; MCI: mild cognitive impairment; CIS: clinically isolated syndrome; DLB: dementia with Lewy bodies; prodDLB: prodromal dementia with Lewy bodies.

## Data Availability

References for this review were identified through searches of Pubmed for articles published from 1997 to 2023.

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
