# Peer review of "Chitinase Signature in the Plasticity of Neurodegenerative Diseases"

_ijms, 2023, doi:10.3390/ijms24076301_

Round 1

Reviewer 1 Report

This review summarizes recent studies of chitinases, in particular CHIT1 and CHI3L1, in several neurodegenerative diseases (NDDs), and discusses the potential of Chitinases as distinctive biomarkers of neuronal cell activity during neuroinflammatory processes. This review is a good summary for people to understand the role of Chitin and Chitnases in NDDs. It also provides guidance for further investigations. I have several comments to improve the manuscript.

1. Figure 1: Why are there arrows in different colors? Authors shall explain the color code, if apply.

2. Some of the statements are not clear, since authors often use “modulate” (eg, line 138-139, line 220), “is related to” (line 208), “correlates with” (eg, line 147-148), and “is involved in” (line 333). Authors may clarify the clear role of chitinases if apply: do they inhibit or activate the protein activities/pathways?

3. Section 3 (line 149-193): The content does not fit the title. The title of this section is “Chitinases and activated microglia in AD”, but the authors mainly discussed microglia as immunomudulatory cells, without any AD context. In addition, authors mentioned chitinases very briefly in 1 or 2 sentenses (IC binding to receptors on microglia, and M2a-microglia induce chitinase 3-like 3, without further details. I think some details shall be provided/discussed to fit the title: eg, how do microglia recognise IC? What's the role of chitinase 3-like 3 in activated microglia during AD?

4. Some sentences are repeated and one of the two can be removed: eg,

“CHI3L1 is modulated by IL-6, IFN-γ, IL-1β and TNF-α. (line 144-145) ”  vs “The expression of CHI3L1 is modulated by pro-inflammatory cytokines, including IL-6, IFN-γ, IL-146 1β and TNF-α. (line 146-147)”; 

line 333-338 vs line 369-373 (CHI3L1 levels in CSF from PD patients).

5. Line 351-354: “Therefore, neuronal αSyn, if it is in excess, is transferred and accumulates in glia, becoming an important stimulus of astroglial inflammatory responses inducing the formation of pathological inclusions and degenerative changes [84].” Besides inflammatory changes, αSyn accumulation in astrocytes causes autophagy defects, resulting in more inclusions. See PMID: 30675347 as a reference.

6. Section 7: Chitinases and Amyotrophic Lateral Sclerosis (ALS): besides TDP-43, there are several other commonly known ALS genes, such as C9RF72, SOD1, FUS, etc. This shall be briefly mentioned.

7. Line 455-457: It's confusing: are you referring to “CHI3L1 and CHI3L2 in CSF” or  “CHI3L1 and CHI3L2 in motor cortex”?

8. Some citations are missing, eg, line 377-380. Proper references shall be cited.

9. Line 197: The form of GlcNAc shall be clarified: GlcNAc polymers?

10. Line 49-51: “These enzymes hydrolase the glycosidic bond of chitin, an important structural constituent of fungi, nematodes, and arthropods [5]. ” hydrolase or hydrolyse?

11. Line 116-117: “κ-light chain of enhancer-activated B cells (NF-κB)” shall be “Nuclear factor κ-light chain of enhancer-activated B cells (NF-κB)”.

Author Response

Dear reviewer,

First of all, we would like to thank for reviewing the paper and suggesting further improvements.

Following your suggestions.

Q1. Figure 1: Why are there arrows in different colors? Authors shall explain the color code, if apply.

R1. The explanation of the arrows different colors has been added in the legend of the figure 1. Thanks a lot.

Q2. Some of the statements are not clear, since authors often use “modulate” (eg, line 138-139, line 220), “is related to” (line 208), “correlates with” (eg, line 147-148), and “is involved in” (line 333). Authors may clarify the clear role of chitinases if apply: do they inhibit or activate the protein activities/pathways?

R2. Sentences in line 208: “It was previously reported that CHIT-1 activity is related to O2 generation and cytokine production in macrophages of peripheral blood of AD and cerebrovascular dementia patients” has been replaced with: “It was previously reported that O2 generation and cytokine production in macrophages of peripheral blood of AD and cerebrovascular dementia patients promote CHIT-1 activity”. The term “modulate” ( line 220) has been replaced with “promote”; the term “correlates (line 147-148) has been changed with induced; the term “is involved in” (line 333) has been replaced with “facilitates”.

Q3. Section 3 (line 149-193): The content does not fit the title. The title of this section is “Chitinases and activated microglia in AD”, but the authors mainly discussed microglia as immunomudulatory cells, without any AD context. In addition, authors mentioned chitinases very briefly in 1 or 2 sentenses (IC binding to receptors on microglia, and M2a-microglia induce chitinase 3-like 3, without further details. I think some details shall be provided/discussed to fit the title: eg, how do microglia recognise IC? What's the role of chitinase 3-like 3 in activated microglia during AD?

R3. Section 3: The title of section 3 has been corrected in “Chitinases and activated microglia in neuroinflammation”.

The details of how IC binds to receptors on microglia, were discussed in the section 2.

Q4. Some sentences are repeated and one of the two can be removed: eg, “CHI3L1 is modulated by IL-6, IFN-γ, IL-1β and TNF-α. (line 144-145)” vs “The expression of CHI3L1 is modulated by pro-inflammatory cytokines, including IL-6, IFN-γ, IL-146 1β and TNF-α. (line 146-147)”; line 333-338 vs line 369-373 (CHI3L1 levels in CSF from PD patients).

R4. The repeated sentences indicated have been fixed.

Q5. Line 351-354: “Therefore, neuronal αSyn, if it is in excess, is transferred and accumulates in glia, becoming an important stimulus of astroglial inflammatory responses inducing the formation of pathological inclusions and degenerative changes [84].” Besides inflammatory changes, αSyn accumulation in astrocytes causes autophagy defects, resulting in more inclusions. See PMID: 30675347 as a reference

R5. Line 351-354: we added the sentence, as you suggested. Thank you.

Q6. Section 7: Chitinases and Amyotrophic Lateral Sclerosis (ALS): besides TDP-43, there are several other commonly known ALS genes, such as C9RF72, SOD1, FUS, etc. This shall be briefly mentioned.

R6. Section 7: The genes, C9RF72, SOD1, FUS, involved in ALS pathogenesis have been briefly mentioned as suggested, thank you.

Q7. Line 455-457: It's confusing: are you referring to “CHI3L1 and CHI3L2 in CSF” or “CHI3L1 and CHI3L2 in motor cortex”?

R7. Line 455-457: the sentence has been corrected.

Q8. Some citations are missing, eg, line 377-380. Proper references shall be cited.

R8. Line 377-380 the reference has been added.

Q9. Line 197: The form of GlcNAc shall be clarified: GlcNAc polymers?

R9. Line 197: GlcNAc has been clarified.

Q10. Line 49-51: “These enzymes hydrolase the glycosidic bond of chitin, an important structural constituent of fungi, nematodes, and arthropods [5]. ” hydrolase or hydrolyse?

R10. Line 49-51: hydrolase has been replaced with hydrolyse.

Q11. Line 116-117: “κ-light chain of enhancer-activated B cells (NF-κB)” shall be “Nuclear factor κ-light chain of enhancer-activated B cells (NF-κB)”.

R11. Line 116-117: “κ-light chain of enhancer-activated B cells (NF-κB)” has been replaced with “Nuclear factor κ-light chain of enhancer-activated B cells (NF-κB)”.

Reviewer 2 Report

The manuscript is scientifically sound. It has potentially high interest to readers interested in the area of science. I suggest the authors consider the following points as they revise their manuscript and continue their work in this important (and attractive) research area.

My specific comments are mentioned below. 

1. The author could check recently published articles.

2. please increase the quality of the figures.

3. increase the text size in both figures.

4. include one paragraph about future aspects.

Author Response

Dear Reviewer,

First of all, we would like to thank for reviewing the paper and suggesting further improvements.

Following your suggestions.

Q1. The author could check recently published articles.

R1. Some of the references reported in this review are recently published articles (2023) as you suggested, thank you.

Q2. Please increase the quality of the figures.

R2. The quality of the figures has been increased. Thank you.

Q3. Increase the text size in both figures.

R3. Done, thank you.

Q4. Include one paragraph about future aspects.

R4. The paragraph has been added, thank you.
